# Diagnostic accuracy of combined thoracic and cardiac sonography for the diagnosis of pulmonary embolism: A systematic review and meta-analysis

Jacqueline Kagima[1,2]*, Marie Stolbrink[1,3], Sheila Masheti[4◉], Collins Mbaiyani[4◉], Aziz Munubi[4,5◉], Elizabeth Joekes[6], Kevin Mortimer[1,3], Jamie Rylance[1,7], Ben Morton[1,3,7]

1 Department of Clinical Sciences, Liverpool School of Tropical Medicine, Liverpool, United Kingdom, 2 Department of Internal Medicine, Kenyatta National Hospital, Nairobi, Kenya, 3 Aintree University Hospital NHS Foundation Trust, Liverpool, United Kingdom, 4 Kenya Medical Research Institute, Centre for Respiratory Disease Research, Nairobi, Kenya, 5 Department of Health Management and Informatics, Kenyatta University, Nairobi, Kenya, 6 Department of Radiology, Royal Liverpool University Hospitals NHS Trust, Liverpool, United Kingdom, 7 Lung Health Group, Malawi-Liverpool-Wellcome Programme (MLW), Blantyre, Malawi

◉ These authors contributed equally to this work.
* jacikagima@gmail.com

## Abstract

### Objectives

Computed tomography pulmonary angiography (CTPA) is the diagnostic standard for pulmonary embolism (PE), but is unavailable in many low resource settings. We evaluated the evidence for point of care ultrasound as an alternative diagnostic.

### Methods

Using a PROSPERO-registered, protocol-driven strategy (https://www.crd.york.ac.uk/PROSPERO, ID = CRD42018099925), we searched MEDLINE, EMBASE, and CINHAL for observational and clinical trials of cardiopulmonary ultrasound (CPUS) for PE. We included English-language studies of adult patients with acute breathlessness, reported according to PRISMA guidelines published in the last two decades (January 2000 to February 2020). The primary outcome was diagnostic accuracy of CPUS compared to reference standard CTPA for detection of PE in acutely breathless adults.

### Results

We identified 260 unique publications of which twelve met all inclusion criteria. Of these, seven studies (N = 3872) were suitable for inclusion in our meta-analysis for diagnostic accuracy (two using CTPA and five using clinically derived diagnosis criterion). Meta-analysis of data demonstrated that using cardiopulmonary ultrasound (CPUS) was 91% sensitive and 81% specific for pulmonary embolism diagnosis compared to diagnostic standard CTPA. When compared to clinically derived diagnosis criterion, CPUS was 52% sensitive

**Data Availability Statement:** All relevant data are within the manuscript and its Supporting Information files.

**Funding:** This research was funded by the National Institute for Health Research (NIHR) (IMPALA, grant reference 16/136/35) using UK aid from the UK Government to support global health research. The views expressed in this publication are those of the author(s) and not necessarily those of the NIHR or the UK Department of Health and Social Care. The corresponding author had full access to all the data in the study and had final responsibility for the decision to submit for publication.

**Competing interests:** The authors declare no conflict of interest. The funders had no role in the design of the study; in the collection, analyses, or interpretation of data; in the writing of the manuscript, and in the decision to publish the results.

and 92% specific for PE diagnosis. We observed substantial heterogeneity across studies meeting inclusion criteria ($I^2 = 73.5\%$).

## Conclusions

Cardiopulmonary ultrasound may be useful in areas where CTPA is unavailable or unsuitable. Interpretation is limited by study heterogeneity. Further methodologically rigorous studies comparing CPUS and CTPA are important to inform clinical practice.

## Introduction

Pulmonary embolism (PE) is a significant public health problem with an estimated 10 million cases per year worldwide [1]. Despite advances in diagnosis and therapy, PE is frequently undiagnosed and untreated; especially in low resource settings [2]. Computed tomography pulmonary angiography (CTPA) is currently considered the diagnostic standard for the diagnosis of PE [3, 4]. However, CTPA is not always available, affordable or feasible, particularly in LMIC settings. A multiorgan ultrasonographic approach has been suggested as a pragmatic alternative diagnostic approach for PE but there is currently a lack of evidence on diagnostic accuracy to inform adoption [5].

A combination of lung ultrasound (LUS) and cardiac echocardiography has been advocated in patients with suspected pulmonary embolism [4, 6]. Ultrasound techniques may be particularly relevant in low-resource settings as this modality is more accessible and less expensive than CTPA. Ultrasound is portable, noninvasive, repeatable, and has no risk of ionizing radiation [7]. Additionally, the use of ultrasound has been shown to promote patient safety [8]. However, use of ultrasonography requires appropriate training and quality assurance. Misinterpretation leads to diagnostic inaccuracy, inappropriate treatment and potential for harm from e.g. unnecessary or delayed anticoagulation. Important barriers to the widespread uptake of ultrasonography include lack of training and a limited evidence base [9].

Our aim was to determine the diagnostic accuracy of cardiopulmonary point of care ultrasound in the diagnosis of pulmonary embolism in acutely breathless adults. To do this, we systematically reviewed and synthesized the published literature. There are meta-analyses in the existing literature establishing the accuracy of a single-organ scan (LUS or cardiac echocardiography applied alone), and in combination with other add-on laboratory and imaging tests for PE diagnosis. To date, no study has systematically reviewed the literature for a multi-organ point of care ultrasound that combines LUS and cardiac echocardiography for PE diagnosis in dyspneic adult patients.

## Materials and methods

We followed the preferred reporting items for systematic reviews and meta-analyses (PRISMA) statement [10]. The checklist is provided in the data in S1 Appendix. The review protocol and search strategy were registered with PROSPERO (http://www.crd.york.ac.uk/PROSPERO, ID: CRD42018099925).

### Data sources and search strategy

The initial systematic search of MEDLINE, EMBASE, and CINAHL databases for English language papers was conducted on 6th June 2017 and updated on 28th February 2020. We

included studies published from 1<sup>st</sup> January 2000 until 28<sup>th</sup> February 2020. The bibliography of included studies was searched for any additional relevant titles. Combinations of subject headings, keywords and synonyms used included: breathless* OR dyspn* OR "shortness of breath"; "pulmonary embol*" OR "pulmonary thromboembol*"; sonogr* OR ultraso* OR "point-of-care ultraso*" OR and "bedside ultraso*" (further details provided in the data in S2 File).

## Study selection and data extraction

We included prospective and retrospective observational studies and clinical trials that recruited acutely breathless adults ($\geq$ 18 years). Non-English papers, unpublished research, conference communications, pediatric studies and studies on animals were excluded. We considered CTPA as the diagnostic standard for PE, but we also included studies that incorporated clinically derived diagnosis as a reference standard. Data was extracted on the study setting, sampling methods, characteristics of the study design, reference standard used in the diagnosis of PE and additional tests done, type of ultrasound machine used, and the organs scanned and sonographers' qualifications, experience and blinding. The diagnostic accuracy measure was our primary outcome measure. Two reviewers (JK and BM) screened titles and abstracts and made study selection decisions independently. Papers that at least one reviewer identified for inclusion were reviewed in full (Fig 1). Data retrieved from these studies by both researchers were compared. Where these were not provided in the original publication, we sent requests for raw data to the authors of studies. Disagreements were resolved by consensus between reviewers (JK and BM) or final arbitration by JR.

## Quality assessment

Methodological quality was assessed through the QUADAS-2 (Quality Assessment of Diagnostic Accuracy Studies) criterion [11], which categorized the risk of bias involving methodology, reporting and validity as low, unclear or high.

## Data analysis

Data were extracted as two-by-two tables (true positives, false positives, true negatives, and false negatives) and analysis conducted using Review Manager 5 (REVMAN) version 5.3 [12] and R-CRAN project version 3.6.3. [13] using the R package 'meta' [14]. Since patients are likely to have more than one abnormal finding per ultrasound, we used the individual patient as the unit of analysis (abnormal finding versus none) and not the individual ultrasound findings.

## Assessment of heterogeneity

Study heterogeneity was evaluated with Cochrane's Q and quantified with the inconsistency ($I^2$) test, using a random effects model. We also assessed heterogeneity by visually inspecting the forest plots to determine closeness of point estimates with each other and overlap of confidence intervals (CIs).

## Publication bias

To assess for publication bias, we used funnel plot [15] for a random-effects model and also performed a regression test with a meta-regression model (Egger's regression test) [16] using R software to assess for plot asymmetry.

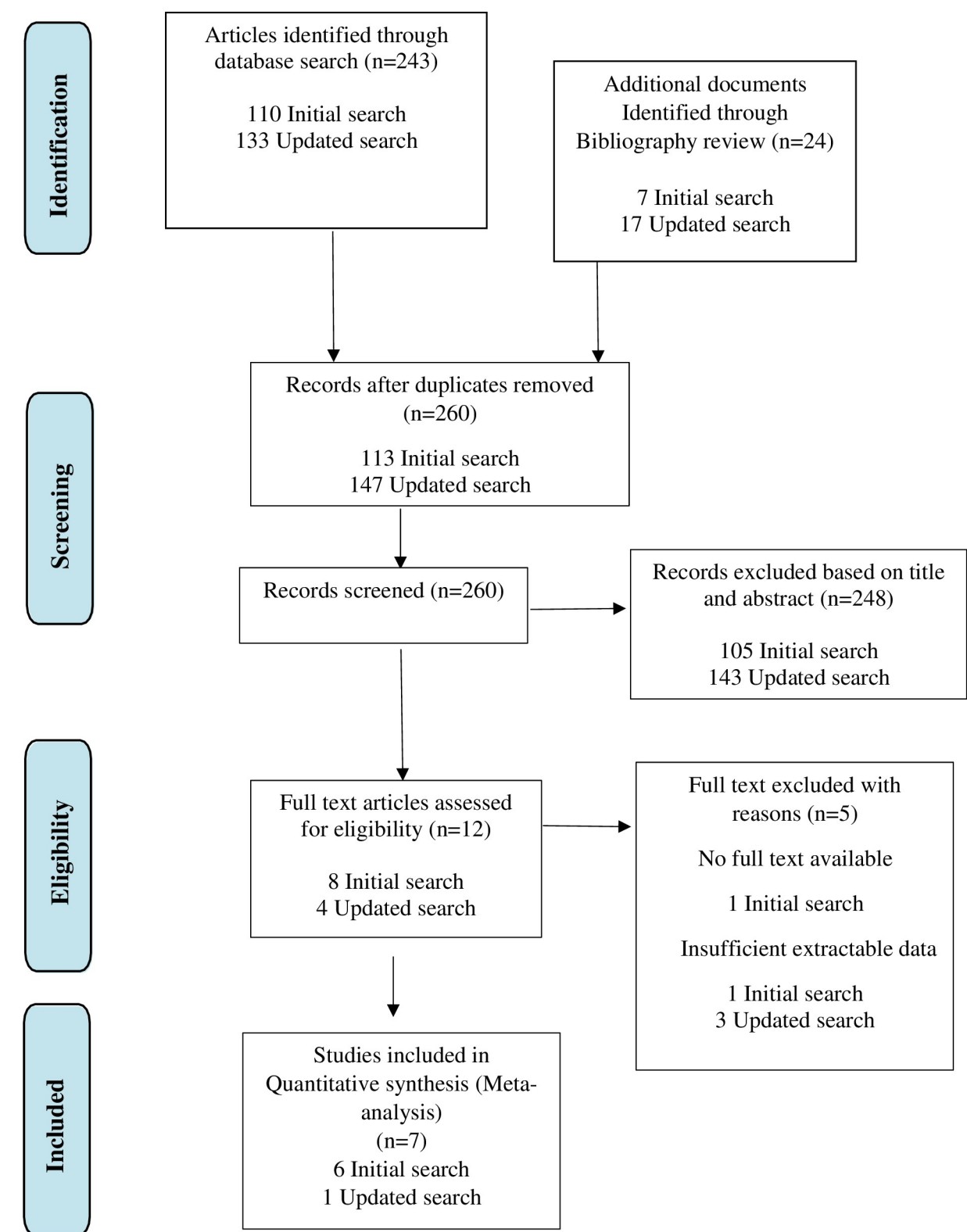

**Fig 1. PRISMA flow diagram showing the study selection process.**

# Results

## Study selection

We identified 260 unique titles of which 248 were excluded during the title and abstract screening process. Following full text review of the remaining 12 articles, five did not meet inclusion criteria for various reasons (see Fig 1 below). Seven articles [17–23] could be included in the meta-analysis, incorporating 3872 patients. Two papers used CTPA as the reference diagnostic standard [19, 20], and five papers used clinically derived diagnosis as the reference standard [17, 18, 21–23] Clinically derived diagnosis adjudicators were blinded from the CPUS results in each of these studies. One study of acutely breathless patients [20] identified PE by exclusion following ultrasound examination for other candidate diagnoses.

## Study characteristics

The main characteristics of eligible studies, which included between 96 and 2683 patients (pooled total 3872) are summarized in Table 1. The majority of the studies were published between 2013 and 2017, and with the exception of one [19] were carried out in single centers [17, 18, 20–23] Six of the included studies were done in emergency departments (ED) and one was done in an intensive care unit (ICU) [21] all in high income country settings (Table 1). All included studies used at least one of the following alternate tests in their clinical diagnosis: blood gases, D-Dimers, chest radiograph (CXR), troponins, brain natriuretic peptide (BNP) or

**Table 1. Baseline characteristics of included studies.**

| Author | Year | Country | Study setting | Sampling | Design | Reference standard | Sample size (n) | Leg US | Ultrasound machine | Sonographer qualifications |
|---|---|---|---|---|---|---|---|---|---|---|
| Laursen [17] | 2013 | Denmark | Single ED | Convenient | Prospective Cohort | Clinically derived diagnosis | 134 | Yes | Vivid S6 GE | 1 Dedicated physician; Done >100 FATE and LCUS scans; >300 FLUS scans |
| Laursen [18] | 2014 | Denmark | Single ED | Consecutive | RCT | Clinically derived diagnosis | 158 | Yes | Logiq S8 GE | 1 Dedicated physician; Done >200 CPUS scans. |
| Bataille [21] | 2014 | France | Single ICU | Consecutive | Prospective Cohort | Clinically derived diagnosis | 136 | No | Sonos 5500 Philips HP | Not reported |
| Nazerian [19] | 2014 | Italy | Multiple ED | Consecutive | Prospective Cohort | CTPA | 357 | Yes | 3 MyLab30 Gold, 1 MyLab40 (Esaote SpA); 1 Logiq3, GE; 1 HD7 (Philips) | 9 EPs with at least 5 years' experience and 4 residents (2 IM and 2EM.) |
| Koenig [20] | 2014 | USA | Single ED | Convenient | Prospective Cohort | CTPA | 96 | Yes | M-turbo Sonosite | 3 pulmonary critical care faculty and 3 third-year fellows. |
| Zanobetti [22] | 2017 | Italy | Single ED | Consecutive | Prospective Cohort | Clinically derived diagnosis | 2683 | No | MyLab 30 Gold Esaote SpA | 10 EPs having attended 80 hours course and done 150 CPUS with 2 years' experience. |
| Jung [23] | 2017 | Korea | Single ED | Consecutive | Prospective Cohort | Clinically derived diagnosis | 308 | No | ACUSON X500 Siemens | Board certified EPs with > 5 years' experience in CPUS and 2 senior EM residents. |

CPUS, cardiopulmonary ultrasound; CTPA, computed tomography pulmonary angiography; ED, emergency department; EM, emergency medicine; EP, emergency physician; FATE, focus-assessed transthoracic echocardiography; FLUS, focused lung ultrasound; IM, internal medicine; ICU, intensive care unit; LCUS, leg compression ultrasound; RCT, randomized controlled study; US, ultrasound.

computed tomography (CT) chest. In all studies the sonographers were blinded to the laboratory results, although these were available to inform clinically derived diagnosis.

Five studies [18, 19, 21–23] enrolled a consecutive sample, and two studies [17, 20] enrolled a convenience sample (Table 1). Only one study included two study arms (randomized control design) [18], of which only the intervention arm included CPUS; we excluded those patients allocated to the control arm. In six studies [17–20, 22, 23], the sonographer was an ED physician, pulmonary or critical care faculty, fellow or resident. The minimum ultrasound experience level varied markedly between studies, ranging from at least two years' experience to more than 10 years' experience. In three studies, the qualification of the sonographer was reported in terms of the minimum number of ultrasound procedures performed, which ranged from 100 to more than 400 [17, 18, 22]. Sonographer experience was not described in one study [21].

Five studies [17, 18, 21–23] used clinically derived diagnosis established by one or more independent adjudicators as the reference standard. In four of these studies [17, 18, 21, 23], the clinical experts who adjudicated the final diagnosis were blinded to the ultrasound findings; however, in one study [22], these clinicians were not blinded to the ultrasound results. Two studies considered as the reference standard exclusively the CTPA scan [19, 20].

## Assessment of methodological quality

**Risk of bias and applicability concerns within studies.**   The quality of all studies was generally high, had low risk of bias, and satisfied the majority of the risk of bias and applicability criteria. Quality assessment of individual studies is summarized in Figs 2 and 3 below.

## Sensitivity and specificity reported by individual studies

The sensitivity for the studies that used a CTPA reference standard ranged from 17% (95% CI = 2%-48%) to 90% (95% CI = 83%-95%), with specificity from 86% (95% CI = 81%-90%) to 100% (95% CI = 96%-100%). For those studies that used clinically derived diagnosis as reference standard, sensitivity ranged from 38% (95% CI = 14%-68%) to 100% (95% CI = 40%-100%), with specificity from 95% (95% CI = 91%-98%) to 100% (95% CI = 99%-100%) (Fig 4 below).

## Synthesis of results

The pooled estimate of the sensitivity of CPUS for detection of PE based on CTPA as the reference standard was 91% (95% CI = 84%-95%) and specificity was 81% (95% CI = 77%-85%). The pooled sensitivity and specificity for detection of PE using clinically derived diagnosis as the reference standard was 52% (95% CI = 43%-61%) and 99% (95% CI = 99%-99%) respectively (Fig 5 below). The pooled diagnostic log odds ratio was 4.72 (95% CI:3.5 to 5.86) as shown on Fig 6 below.

## Heterogeneity assessment

Significant heterogeneity was found across studies (Q (df = 6) = 18.0788, p-value = 0.006), of moderate magnitude ($I^2$ = 73.5%). Total heterogeneity, as estimated by $tau^2$, was 1.4 (95% CI:0.11 to 14.15). Studies by Bataille (performed in ICU) [21] and Jung (in ED) [23] contributed most to both overall heterogeneity and results as demonstrated by the Baujat plot [24] below (Fig 7). The former study did not indicate the qualifications of the sonographer.

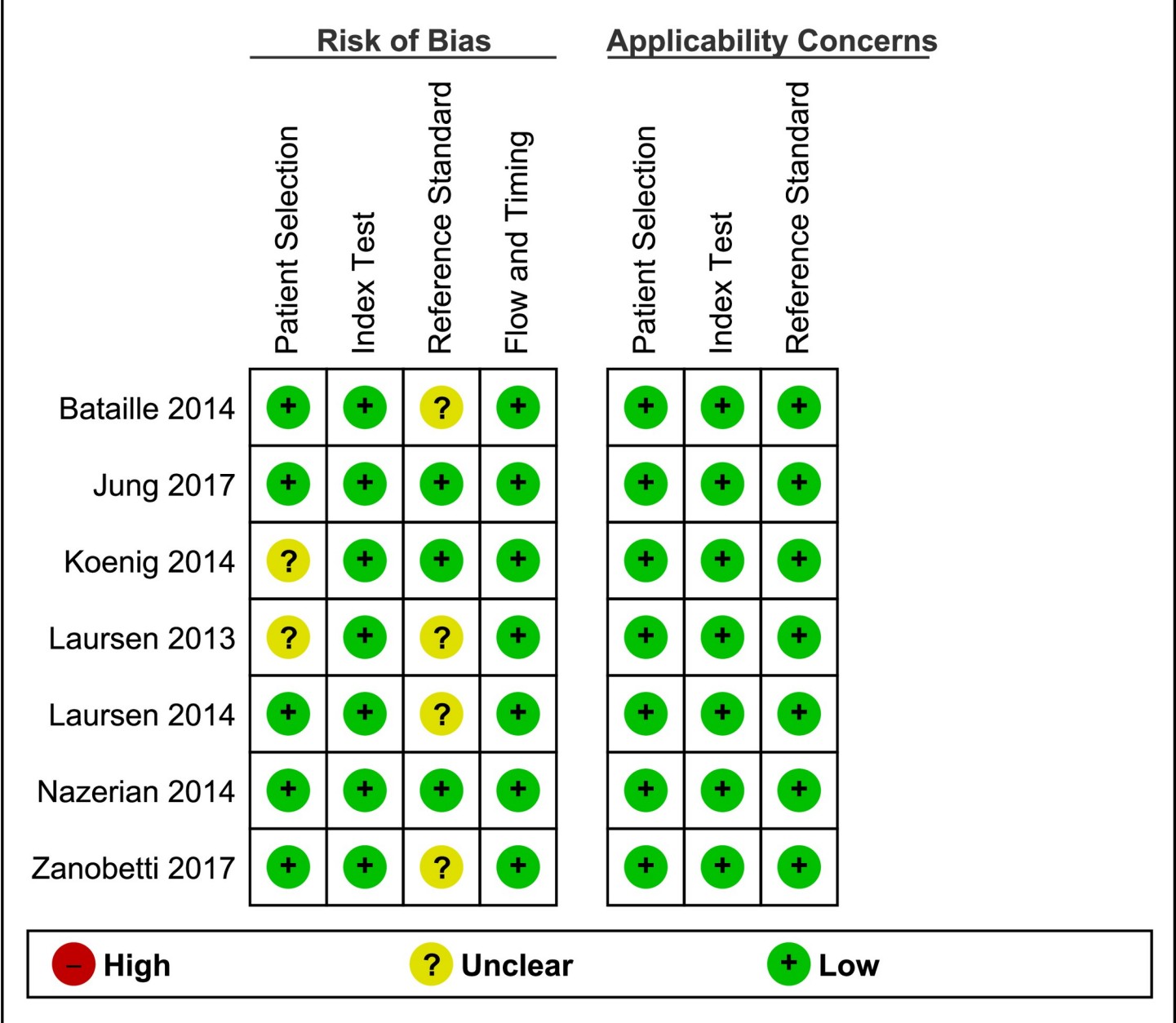

**Fig 2. Risk of bias and applicability concerns summary.** Review of authors' judgments about each domain for each included study. Majority of the studies had a low risk of bias.

## Publication bias

The funnel plot for a model without moderators (random-effects model), demonstrated symmetry, and no evidence of publication bias (Fig 8), and Egger's regression test [16] for plot asymmetry was consistent with this (p = 0.21).

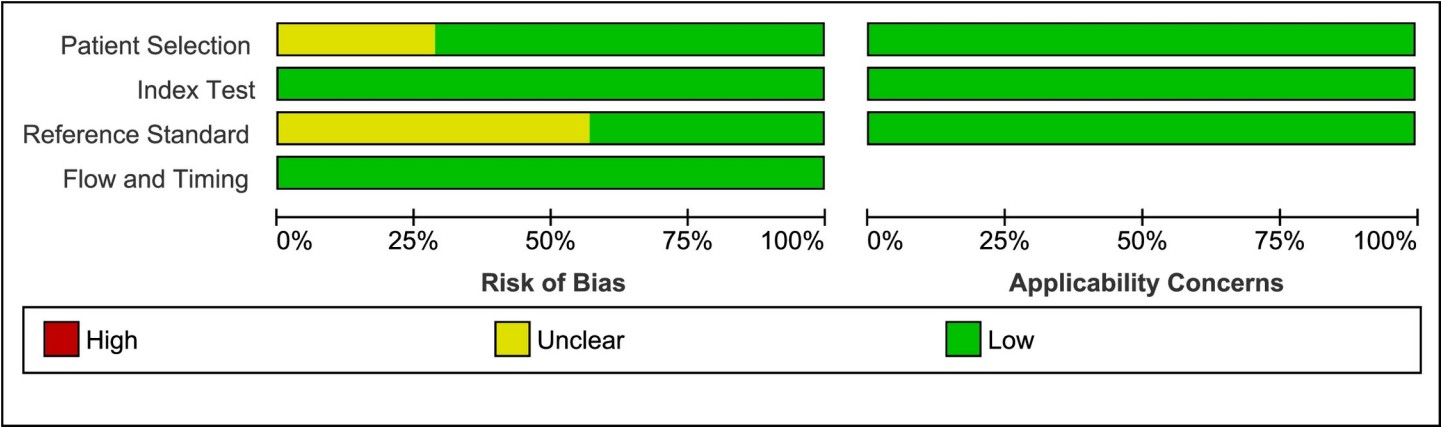

**Fig 3. Risk of bias and applicability concerns graph.** Review of authors' judgments about each domain presented as percentages across included studies Majority of the studies had a low risk of bias.

## Discussion

The findings of this systematic review suggest that cardiopulmonary ultrasound could play an important role in PE diagnostics. In this meta-analysis we found that cardiopulmonary ultrasound (CPUS) is 91% sensitive and 81% specific for a diagnosis of PE when compared to the diagnostic standard CTPA. We identified a number of studies that compared ultrasound with clinically derived PE diagnosis; interpretation of such studies, where the clinical reference standard is known to be poorly sensitive and specific compared to CTPA, is challenging. Optimal PE diagnostics may not be feasible in specific settings such as when CTPA is unavailable or patient transfer is unsafe, and it appears that the CPUS may offer a viable alternative.

Previous metanalyses found that the sensitivities of single organ ultrasonographic test were lower than those of CPUS. Echocardiography used as a single ultrasonographic test in PE diagnostics in critically ill adults showed a 53% sensitivity and 83% specificity in a recent meta-analysis [25]. LUS used as a single ultrasonographic test to diagnose PE in adults admitted to EDs and medical wards showed an 87% sensitivity and 81.8% specificity [26]. These results

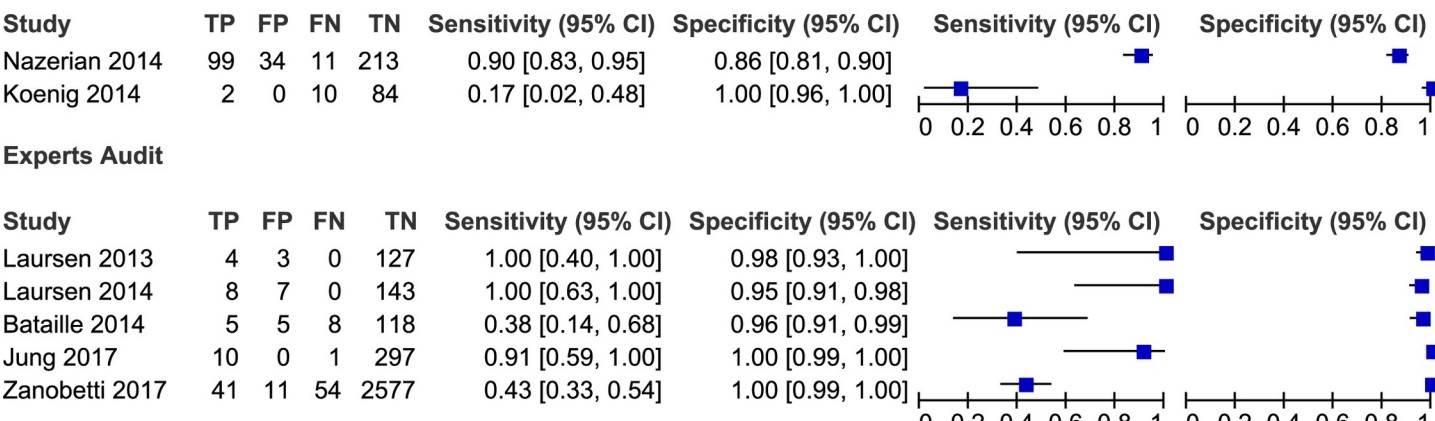

**Fig 4. Forest plot of sensitivity and specificity of the individual studies.** Forest plots presenting the estimates of sensitivity and specificity and 95% credibility intervals of each study across the two reference standards used (CTPA and clinically derived diagnosis). FN, false negative; FP, false positive; TN, true negative; TP, true positive.

**Pooled CTPA**

| Study | TP | FP | FN | TN | Sensitivity (95% CI) | Specificity (95% CI) |
|---|---|---|---|---|---|---|
| Pooled data | 111 | 62 | 11 | 269 | 0.91 [0.84, 0.95] | 0.81 [0.77, 0.85] |

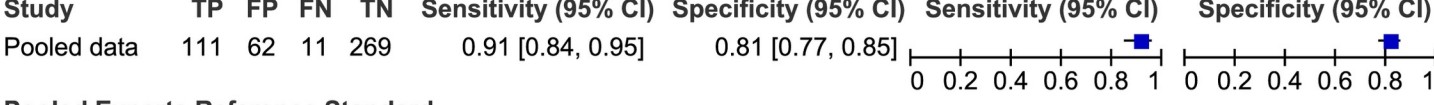

**Pooled Experts Reference Standard**

| Study | TP | FP | FN | TN | Sensitivity (95% CI) | Specificity (95% CI) |
|---|---|---|---|---|---|---|
| Pooled data | 68 | 26 | 63 | 3262 | 0.52 [0.43, 0.61] | 0.99 [0.99, 0.99] |

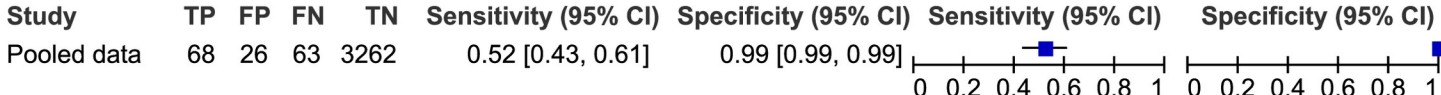

**Fig 5. Forest plot of pooled estimates of sensitivity and specificity of the reference standard modalities.** Forest plots presenting the pooled estimates of sensitivity and specificity and 95% credibility intervals of the two reference standards used (CTPA and clinically derived diagnosis). FN, false negative; FP, false positive; TN, true negative; TP, true positive.

confirm the limitation of a single-organ ultrasonography to rule out PE. The Bedside Lung Ultrasound in Emergency (BLUE) protocol is a protocol that allows diagnosis of PE based on a positive vein ultrasonography in combination with the exclusion by lung ultrasonography of other causes of respiratory failure, introduced by Lichtenstein and Mezière. The BLUE protocol showed a sensitivity of 81% and a specificity of 99% for PE diagnostics [27]. BLUE protocol has also been tested in combination with echocardiography in a small study performed in ED patients. None of the patients with an alternative ultrasonographic finding had PE on CTPA, thus showing an added value of CPUS in PE diagnostics [20]. These studies show a consistently higher specificity compared to the sensitivity and this could be due to the inability to rule out PE without the CTPA. Nevertheless, these findings suggested that the use of ultrasound has the potential to reduce the number of unnecessary CTPAs [6].

As with any ultrasonographic test, ultrasound is operator dependent. Specific training is required, and the experience of the sonographer is necessary to increase test accuracy. There

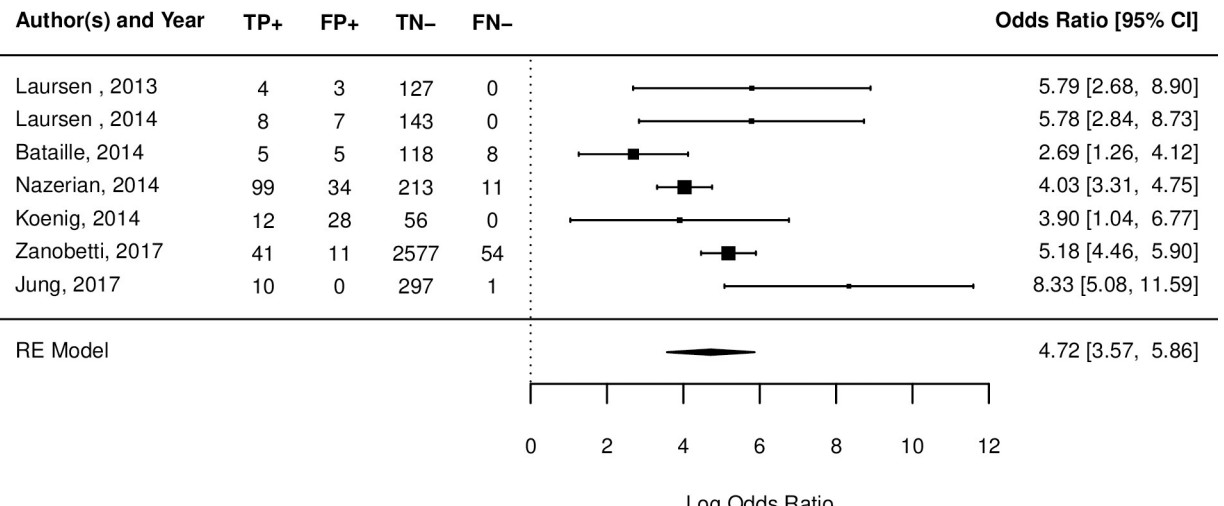

**Fig 6. Forest plot of pooled diagnostic odds ratio of CPUS in diagnosing PE.** Forest plots presenting the odds ratios and 95% credibility intervals of all the studies (FN, false negative; FP, false positive; TN, true negative; TP, true positive.

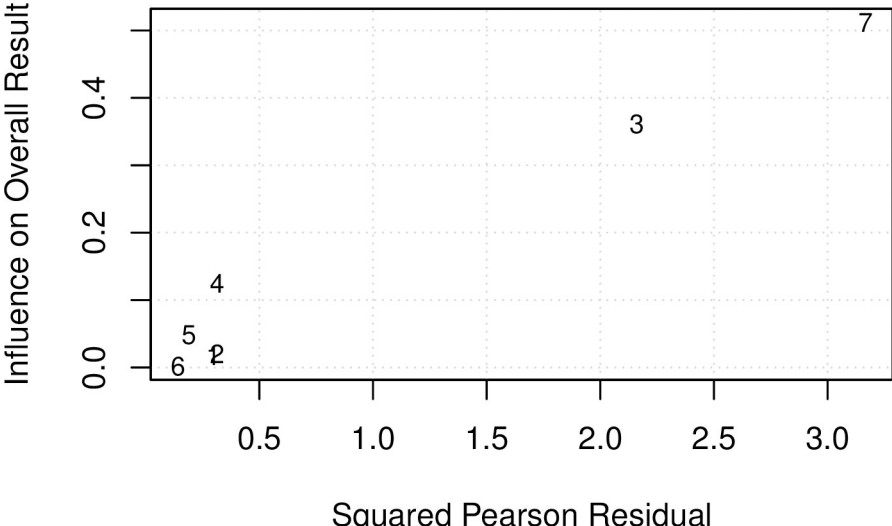

**Fig 7. Baujat plot.** 1-Laursen 2013, 2-Laursen 2014, 3- Bataille, 4- Nazerian, 5- Koenig, 6-Zanobetti, 7-Jung. Studies that fall to the top right quadrant of the Baujat plot contribute most to both the overall heterogeneity and overall results.

are often incidental findings on CTPA which may or may not have clinical significance, but which increase anxiety, use of further diagnostic tests and consume clinical and patient time and resources [28]. The European Society of Cardiology (ESC) suggests that the best diagnostic strategy to confirm or exclude PE is to combine clinical assessment, plasma D-dimer measurement and computed tomographic pulmonary angiography (CTPA) [29]. As with all diagnostic tests, the accuracy of each diagnostic method is highly enhanced when used in conjunction with others. Considering this, a multi-organ ultrasound approach 'triple scan' has been

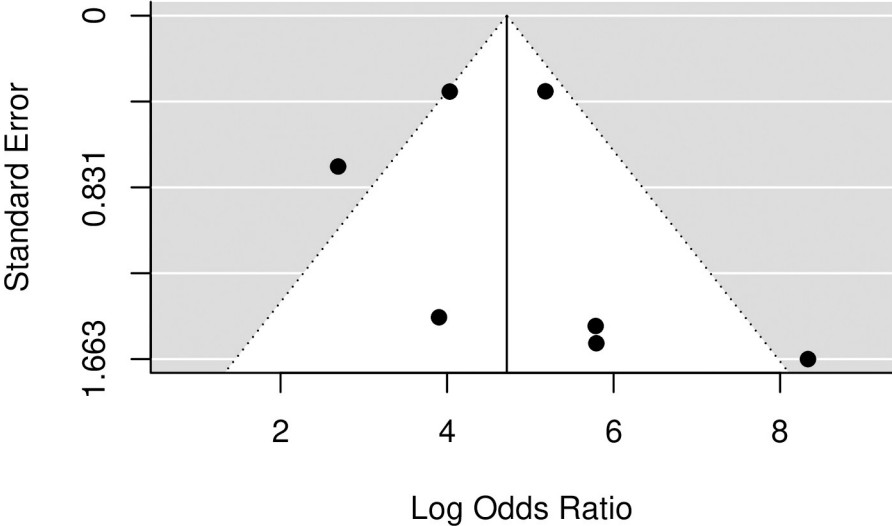

**Fig 8. Funnel plot for the random effects model.** Plot shows symmetry.

proposed which includes the use of cardiac, lung, and leg compression ultrasound scan. A proposed algorithm for the diagnosis of PE when other imaging diagnostics are not available is included in the data in S2 Fig. However, there is no robust evidence for the superiority of this approach currently. Nevertheless, our review highlights the promise of CPUS for evaluating patients suspected of having PE.

The strengths of our systematic review and meta-analysis include its comprehensive scope and search and the quality of the studies was generally high. All studies ensured proper blinding of the sonographer to the diagnostic test results. Limitations include the moderate heterogeneity between the included studies which puts some caution to the interpretation of these results. These data reflect the methodological diversity of the analyzed studies, as described previously (study setting, design, sampling, blinding, sonographer's expertise, and reference standard). There are no universal standardized approaches to US delivery and reporting and this lack of core outcomes sets (COS) contributes to the heterogeneity between trials [30]. We also recognize that variability was inherent in this meta-analysis because bedside lung or cardiac ultrasound findings are not pathognomonic for PE. Lastly, clinically derived PE diagnosis is a subjective reference standard [31]. Subjectivity in the studies we included was minimized by using a predefined protocol for most studies [17, 18, 22, 23] based on internationally accepted diagnostic criteria. Since patients are likely to have more than one abnormal finding per ultrasound, we used the individual patient as the unit of analysis (abnormal finding versus none) and not the individual ultrasound findings. Clinically, it is also useful to know the accuracy of individual ultrasound findings as it is plausible that some findings are better indicators of PE than others.

## Conclusions

Our findings suggest that cardiopulmonary ultrasonography has potential to be used in PE diagnostics when CTPA is unsuitable or unavailable. CPUS may be highly relevant in low resource settings as it is cheaper and is less human resource dependent compared to CTPA. Based on the current evidence, a global harmonization of US diagnostic criteria for PE is required, potentially within worldwide consensus guidelines that will allow specifications of essential practice standards for low resource settings. Given the complexity of clinical investigation, future studies should investigate the impact of a point of care CPUS on diagnostic certainty, wider resource and test use, and the effects on patient pathways and treatment plans.

## Supporting information

**S1 Appendix. Link to protocol.** https://www.crd.york.ac.uk/PROSPERO.
ID = CRD42018099925.
(TXT)

**S1 Fig. Pooled receiver operator characteristic curve of CPUS in diagnosing PE.** Summary receiver operating characteristic (SROC) curve illustrating the pooled sensitivity and specificity of the two reference standards used (CTPA [black] and Experts clinicians audit [red]).
(DOCX)

**S2 Fig. Proposed algorithm for pulmonary embolism diagnosis.** The proposed algorithm that considers the use of the triple point of care ultrasonography of the heart, lung and leg veins for the diagnosis of PE.
(DOCX)

**S1 File. PRISMA checklist.**
(DOC)

**S2 File. Search strategies and search terms.**
(DOCX)

## Acknowledgments

The authors thank Chris Jewell (University of Lancaster, UK) for his statistical input.

## Author Contributions

**Conceptualization:** Jacqueline Kagima, Ben Morton.

**Data curation:** Jacqueline Kagima, Sheila Masheti, Collins Mbaiyani, Aziz Munubi, Ben Morton.

**Formal analysis:** Jacqueline Kagima, Jamie Rylance, Ben Morton.

**Funding acquisition:** Kevin Mortimer.

**Investigation:** Jacqueline Kagima, Sheila Masheti, Collins Mbaiyani, Aziz Munubi, Jamie Rylance, Ben Morton.

**Methodology:** Jacqueline Kagima, Jamie Rylance, Ben Morton.

**Project administration:** Jacqueline Kagima.

**Resources:** Jacqueline Kagima, Marie Stolbrink, Elizabeth Joekes, Ben Morton.

**Software:** Jacqueline Kagima, Marie Stolbrink, Jamie Rylance.

**Supervision:** Kevin Mortimer, Jamie Rylance, Ben Morton.

**Validation:** Jacqueline Kagima, Jamie Rylance, Ben Morton.

**Visualization:** Jacqueline Kagima, Marie Stolbrink, Jamie Rylance, Ben Morton.

**Writing – original draft:** Jacqueline Kagima.

**Writing – review & editing:** Jacqueline Kagima, Marie Stolbrink, Elizabeth Joekes, Kevin Mortimer, Jamie Rylance, Ben Morton.

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
