## [Decision Letter · Decision Letter 0]

14 Jul 2020

PONE-D-20-19213

Diagnostic accuracy of combined thoracic and cardiac sonography for the diagnosis of pulmonary embolism: a systematic review and meta-analysis

PLOS ONE

Dear Dr. Kagima,

Thank you for submitting your manuscript to PLOS ONE. After careful consideration, we feel that it has merit but does not fully meet PLOS ONE’s publication criteria as it currently stands. Therefore, we invite you to submit a revised version of the manuscript that addresses the points raised during the review process.

 I have received the comments of the reviewers on your manuscript. The specific comments of the reviewers are included below. Please provide point by point response in your revised manuscript.

We look forward to receiving your revised manuscript.

Kind regards,

Muhammad Adrish

Academic Editor

PLOS ONE

Journal Requirements:

2. At this time, we ask that you please provide the full search strategy and search terms for at least one database used as Supplementary Information.

3. In the Methods section, please provide the specific information extracted from the articles selected in your study in the 'Study selection and data extraction sub-section.

Reviewers' comments:

Reviewer's Responses to Questions

**Comments to the Author**

1. Is the manuscript technically sound, and do the data support the conclusions?

Reviewer #1: Yes

Reviewer #2: Yes

2. Has the statistical analysis been performed appropriately and rigorously? 

Reviewer #1: N/A

Reviewer #2: Yes

3. Have the authors made all data underlying the findings in their manuscript fully available?

Reviewer #1: Yes

Reviewer #2: Yes

4. Is the manuscript presented in an intelligible fashion and written in standard English?

Reviewer #1: Yes

Reviewer #2: Yes

5. Review Comments to the Author

Reviewer #1: An interesting manuscript in its field, I suggest that the authors add a figure with their proposed algorithm when all tools for the diagnosis of pulmonary embolism are not available. Otherwise I do not want any other corrections

Reviewer #2: the authors conducted a meta analysis of an important clinical problem -i.e., to look at validity of ultrasound to diagnose pulmonary embolism using two different reference criteria; the stronger one - CT chest angiography - being used in only two studies, unfortunately; the outcome is very helpful to address this difficult diagnosis in low resource settings where CT is perhaps not available, while ultrasound of heart, lungs and legs combined appear very helpful to address this diagnostic dilemma. The study team used all standard ways to do data extraction, perform quality assessment of included studies, looking at forest plots, funnel plots to explore possible publication bias, using standard reporting and having registered the study at PROSPERO, all meeting the criteria for high quality.

6. PLOS authors have the option to publish the peer review history of their article (what does this mean?). If published, this will include your full peer review and any attached files.

Reviewer #1: No

Reviewer #2: No

---

## [Author Response · Author response to Decision Letter 0]

27 Aug 2020

RESPONSE TO REVIEWER 1;

Response to Reviewer 1: An interesting manuscript in its field, I suggest that the authors add a figure with their proposed algorithm when all tools for the diagnosis of pulmonary embolism are not available. 

Thank you for your review of our paper and comment. We agree and have added a proposed algorithm in our discussion (page 16, line 264-265) and added the figure under supporting information (S2 Fig). 

RESPONSE TO REVIEWER 2;

The authors conducted a meta-analysis of an important clinical problem -i.e., to look at validity of ultrasound to diagnose pulmonary embolism using two different reference criteria; the stronger one - CT chest angiography - being used in only two studies, unfortunately; the outcome is very helpful to address this difficult diagnosis in low resource settings where CT is perhaps not available, while ultrasound of heart, lungs and legs combined appear very helpful to address this diagnostic dilemma. The study team used all standard ways to do data extraction, perform quality assessment of included studies, looking at forest plots, funnel plots to explore possible publication bias, using standard reporting and having registered the study at PROSPERO, all meeting the criteria for high quality

Thank you for your review of our paper. We appreciate these comments. We agree indeed that ultrasound of the heart, lungs and legs combined may be very helpful in in low resource settings especially when CTPA is not immediately available or feasible.

---

## [Decision Letter · Decision Letter 1]

2 Sep 2020

Diagnostic accuracy of combined thoracic and cardiac sonography for the diagnosis of pulmonary embolism: a systematic review and meta-analysis

PONE-D-20-19213R1

Dear Dr. Kagima,

We’re pleased to inform you that your manuscript has been judged scientifically suitable for publication and will be formally accepted for publication once it meets all outstanding technical requirements.

Kind regards,

Muhammad Adrish

Academic Editor

PLOS ONE

Additional Editor Comments (optional):

Reviewers' comments:

Reviewer's Responses to Questions

**Comments to the Author**

1. If the authors have adequately addressed your comments raised in a previous round of review and you feel that this manuscript is now acceptable for publication, you may indicate that here to bypass the “Comments to the Author” section, enter your conflict of interest statement in the “Confidential to Editor” section, and submit your "Accept" recommendation.

Reviewer #1: All comments have been addressed

2. Is the manuscript technically sound, and do the data support the conclusions?

Reviewer #1: Yes

3. Has the statistical analysis been performed appropriately and rigorously? 

Reviewer #1: Yes

4. Have the authors made all data underlying the findings in their manuscript fully available?

Reviewer #1: Yes

5. Is the manuscript presented in an intelligible fashion and written in standard English?

Reviewer #1: Yes

6. Review Comments to the Author

Reviewer #1: I have no comments, all indications were addressed correctly. I have no further comments. Thank you for your collaboration

7. PLOS authors have the option to publish the peer review history of their article (what does this mean?). If published, this will include your full peer review and any attached files.

Reviewer #1: No

---

## [Editor Report · Acceptance letter]

7 Sep 2020

PONE-D-20-19213R1 

Diagnostic accuracy of combined thoracic and cardiac sonography for the diagnosis of pulmonary embolism: a systematic review and meta-analysis 

Dear Dr. Kagima:

I'm pleased to inform you that your manuscript has been deemed suitable for publication in PLOS ONE. Congratulations! Your manuscript is now with our production department. 

Kind regards, 

on behalf of

Dr. Muhammad Adrish 

Academic Editor

PLOS ONE